# Relationship between Sensitivity Tendency and Psychological Stress Reactivity in Healthy Students

**DOI:** 10.3390/healthcare10050949

**Published:** 2022-05-20

**Authors:** Syunsaku Ishibashi, Jun Murata, Akiko Tokunaga, Akira Imamura, Kojiro Kawano, Ryoichiro Iwanaga, Goro Tanaka

**Affiliations:** 1Department of Occupational Therapy Sciences, Nagasaki University Graduate School of Biomedical Sciences, 1-7-1 Sakamoto, Nagasaki 852-8520, Japan; le.sln.7.lfe.mdc@gmail.com (S.I.); aimamura@nagasaki-u.ac.jp (A.I.); iwanagar@nagasaki-u.ac.jp (R.I.); goro@nagasaki-u.ac.jp (G.T.); 2Saikai Hospital, 1500 Gonjojimachi, Sasebo 859-3213, Japan; 3Department of Health Sciences, Nagasaki University Graduate School of Biomedical Sciences, 1-7-1 Sakamoto, Nagasaki 852-8520, Japan; akiko0923@nagasaki-u.ac.jp; 4Tikumaso Mental Hospital, 4-6 Chuouhigashi, Ueda 386-8584, Japan; kkawano@shinshu-u.ac.jp

**Keywords:** sensory processing sensitivity, highly sensitive persons, galvanic skin response, Stroop task

## Abstract

This study examined the relationship between sensory processing sensitivity and psychological stress reactivity in 69 healthy Japanese university students. The Japanese version of the Highly Sensitive Person Scale and the Japanese version of the Adolescent/Adult Sensory Profile were used for subjective assessment. The Galvanic skin response was measured as an objective measure of stress responses while the participants were completing the Stroop task. The Wilcoxon signed-rank test, the Spearman rank correlation coefficient, and the Mann–Whitney U test were conducted for data analysis. The results demonstrated that there was no significant correlation between the Japanese version of the Highly Sensitive Person Scale and Galvanic skin response. However, there was a marginal trend toward significance between low registration in the Japanese version of the Adolescent/Adult Sensory Profile and Galvanic skin response (r_s_ = 0.231, *p* < 0.10; r_s_ = 0.219, *p* < 0.10), suggesting that self-rated sensitivity was not necessarily associated with objective measures. These results indicate that sensory processing sensitivity analyses require the consideration of the traits and characteristics of the participants and multifaceted evaluations using a sensitivity assessment scale other than the Japanese version of the Highly Sensitive Person Scale.

## 1. Introduction

Sensitivity to the surrounding environment and stimuli varies between people [1,2]. Sensory processing sensitivity (SPS) denotes innate sensitivity and represents individual differences in the course of processing sensory information in the brain rather than the sensory organs themselves [3]. People with high SPS are called highly sensitive persons (HSPs) [4]. HSPs are characterized by deep cognitive processing and high emotional responses to stimuli [5] and account for about 20% of the world population [3,6]. Because of their temperament, HSPs are easily overwhelmed by external stimuli, such as loud noises and strong light [7], and are highly sensitive to internal stimuli, such as their own emotions [8]. As SPS is an innate trait, it is important to help HSPs to recognize it and support them in developing appropriate self-care and coping strategies [9]. However, a quantitative evaluation method for determining the degree of HSPs has not yet been established.

SPS has attracted increasing interest in multiple areas of psychology in recent years [10]. High SPS results from the brain’s ability to process information deeply [11]. The original version of the Highly Sensitive Person Scale (HSPS) was developed as a measure of SPS [3]. The degree of SPS, as measured by the HSPS, is reportedly associated with depression and anxiety [12]; mental health problems, such as fatigue and tension [13]; subjective well-being [14]; and life satisfaction [15]. In recent years, there have been reports on attention deficit hyperactivity disorder [16], seasonal affective disorder [17], obsessive–compulsive disorder [18], depressive symptoms [19], coronavirus disease 2019 (COVID-19)-related stress [20], personality traits [21], burnout [22], and compassion fatigue [23] in studies related to SPS. However, many of these studies’ data lack objectivity because they employ self-assessment methods. In this regard, some previous studies have indicated that evidence using objective measures is scarce [24,25].

Significantly, some previous studies have reported an association between HSPS scores and stronger activation of the emotional regions of the brain (the cingulate gyrus and the insular cortex) [25,26]. Changes in the emotional regions of the brain may be reflected in physiological responses, such as sweating, which is frequently used as an objective measure of mental stress in psychological tests. In addition, another previous study [27] reported that HSPS scores were associated with the scores of the Adolescent/Adult Sensory Profile (AASP), which evaluates the degree of sensory processing patterns by self-assessment methods. The degree of AASP, similar to that of HSPS, may be related to sweating as a response to mental stress. However, no previous studies have investigated whether the degrees of HSPS and AASP are associated with changes in psychological sweating. Therefore, this study hypothesized that a higher degree of subjectively assessed SPS is associated with a stronger sweating response to mental stress. To examine this hypothesis, this study investigated the relationship between sensitivity tendency, including SPS, and the palmar sweating response. The results indicate that self-rated sensitivity might not be associated with an objective measure. They also indicate that participants’ traits and characteristics must be considered and that multifaceted evaluations must be performed using a sensitivity assessment scale other than the Japanese version of the HSPS to conduct a sensory processing sensitivity analysis.

## 2. Materials and Methods

### 2.1. Participants and Procedures

The study was conducted from October 2019 to October 2021. It included 69 healthy participants (32 men and 37 women) with a mean age of 20.49 ± 1.45 years. All participants were Japanese public university students. This study was performed in accordance with the Declaration of Helsinki and approved by the Ethics Committee of the Department of Health, Graduate School of Biomedical Sciences, Nagasaki University (Nagasaki, Japan, approval number: 18071203).

A letter explaining the study was distributed. The students were also informed that participation was voluntary and optional and would not affect their grades. Then, convenience sampling was performed to recruit participants from the acquaintances of the first author and by word of mouth. The purpose and content of this study were fully explained verbally and in writing, and only those who gave their informed consent were included in the survey. In addition, the participants were informed that anonymity would be maintained and their personal information would be protected. They completed the study questionnaires in the presence of an investigator in the university.

### 2.2. Assessment

#### 2.2.1. Japanese Version of the Highly Sensitive Person Scale (HSPS-J19)

The Japanese version of the original HSPS (HSPS-J19) [3] is a self-reported rating scale comprising 19 items on a 7-point scale [28]. The scale has a 3-factor structure, including low sensory threshold (LST; 7 items); ease of excitation (EOE; 8 items), which indicates reactivity to stimuli; and aesthetic sensitivity (AES; 4 items), which indicates mental richness, such as having a rich imagination [29]. Higher total scores indicate higher SPS. AES reflects positive aspects of SPS, such as artistic sensitivity [30], and contains characteristics that are different from those of other subscales [14,31]. In addition, as AES includes items regarding a rich inner life, it may be subject to social desirability [5] and should be interpreted with caution because of its low internal consistency reliability [32]. As the purpose of this study was to examine the relationship between SPS and psychological stress reactivity, this study followed a study that focused on the negative aspects of SPS [33] and included only two subscales, the LST and EOE. Although the reliability and validity of this scale have been demonstrated, correlations with other scales for each factor of this scale require examination [28]. The Cronbach’s alpha coefficients for the participants in this study were LST = 0.861 and EOE = 0.828.

#### 2.2.2. Japanese Version of the Adolescent/Adult Sensory Profile (AASP)

The AASP is a self-reported rating scale that measures the degree of sensory processing patterns comprising 60 items on a 5-point scale [34]. This scale can be classified into four areas: low registration, sensory sensitivity, sensation seeking, and sensation avoiding. Higher scores in each area indicate stronger sensory patterns in that area. Low registration represents high thresholds for and weak responses to sensory stimuli. Sensation seeking represents high thresholds for sensory stimuli and enjoying or pursuing sensory stimuli. Sensory sensitivity represents low thresholds for and discomfort with sensory stimuli. Sensation avoiding is characterized by high reactivity to and avoidance of sensory stimuli as coping behaviors. The reliability and validity of this scale were reported in a previous study [35]. The Cronbach’s alpha coefficients for the participants in the present study were as follows: low registration = 0.736, sensory sensitivity = 0.607, sensation-seeking = 0.664, and sensation avoidance = 0.520.

#### 2.2.3. Galvanic Skin Response

Galvanic skin response (GSR) was used to electrically capture mental sweating. GSR is designed to quantify and graphically represent the changes in the electric potential between electrodes attached to the skin caused by palmar sweating due to psychological agitation, such as tension or stress [36]. GSR values fluctuate in a negative direction when a stress response occurs. GSR electrodes were placed 2 cm below the middle finger MP joint, where sympathetic nerve responses are prominent [37], and on the proximal 1/3 of the forearm length, as previously reported [38]. A ground electrode was placed on the forearm. The GSR signals and the marking signal were simultaneously stored on a computer using an analog–digital converter (ProComp Infiniti-T7500M, Thought Technology, Montreal, QC, Canada) at a sampling frequency of 256 Hz and analyzed using computer software (BioGraph INFINITI, Thought Technology, Montreal, QC, Canada).

#### 2.2.4. Stroop Task

The Stroop task [39] was conducted to measure stress using two types of conditions: the congruent condition, in which the word for a color, such as red or blue, was written in the same color as the color the word represents (e.g., the word “red” written in red), and the incongruent condition, in which the word for a color was written in a different color than the color the word represents (e.g., the word “blue” written in red). The participants were asked to answer with the color of the letters. This test activates the cingulate gyrus [40,41] and the prefrontal cortex [42,43], which are emotional regions of the brain, and it is widely used as a measure of attention and executive control [44].

### 2.3. Experimental Procedure

The experiment was conducted in a room shielded to the greatest extent possible from external sound and light. After all preparations were finished, each participant was allowed to sit in a chair for more than 5 min to stabilize their GSR. The participants were instructed to gaze at the center of a 1.0 cm diameter circle displayed at the center of a computer monitor positioned 100 cm away. After confirming their resting state, the experiment was started.

In the Stroop task, the participants completed both the congruent and incongruent conditions, with 12 patterns for each condition. The computer screen displayed the words “red”, “blue”, “green”, or “yellow” written in Japanese and printed in either red, blue, green, or yellow. The order of the tasks was randomized for each participant. The participants were instructed to respond to the tasks displayed on the monitor as quickly as they could. After they answered correctly, the next task was displayed. They were verbally instructed to keep their body movements to a minimum during the experiment.

### 2.4. Data Analysis

The data of the GSR was averaged every second. The mean value of GSR obtained for 5 s before the start of the Stroop task was defined as the baseline level. In each procedure (the congruent and incongruent conditions), the changes in GSR (%) from the baseline levels for an individual participant were aligned at the onset of the task. The maximum rates of change during the task were calculated and averaged among participants. The maximum change in GSR during the congruent and incongruent conditions was measured, and differences between the two procedures were compared using the Wilcoxon signed-rank test. The Spearman rank correlation coefficient was performed to determine the correlations among the changes in GSR and the HSPS-J19 and AASP scores. Furthermore, to identify the gender effect on each measurement datum, the Mann–Whitney U test was performed to compare the responses in GSR and the HSPS-J19 and AASP scores between males and females. The data were statistically analyzed using SPSS version 22.0 (IBM, New York, NY, USA); the significance level was set at 5%, and the marginal significance level was set at 10%. There were no missing data in this study.

## 3. Results

### 3.1. HSPS-J19 Scores of the Research Participants

The HSPS-J19 scores (mean ± SD) of the 69 participants in this study were 57.49 ± 14.77 for HSPS total, 27.39 ± 8.02 for LST, and 32.43 ± 8.20 for EOE.

### 3.2. Comparison of GSR between the Congruent and Incongruent Conditions in the Stroop Task

The maximum rates of change in GSR during the congruent and incongruent conditions were −33.68 ± 30.48% and −44.85 ± 29.85%, respectively. The GSR response to the incongruent condition in the Stroop task was much smaller than the response to the congruent condition (*p* < 0.001; Table 1).

### 3.3. Relationships between the Scores of HSPS-J19 or AASP and the Maximum Rate of Change in GSR

There were no significant correlations between the scores of the HSPS-J19 or AASP and the maximum rate of change in GSR in the congruent or incongruent conditions (Table 2). As for the AASP, there was a marginal trend toward significance between GSR and low registration in the congruent condition (r_s_ = 0.231, *p* < 0.10) and the incongruent condition (r_s_ = 0.219, *p* < 0.10; Table 3).

### 3.4. Relationship between the Scores of HSPS-J19 and AASP

There were significant correlations between the total HSPS-J19 score and low registration (r_s_ = 0.323, *p* < 0.01), sensory sensitivity (r_s_ = 0.425, *p* < 0.01), and sensation avoiding (r_s_ = 0.434, *p* < 0.01) of the AASP. There were significant correlations between the LST (low sensory threshold) of the HSPS-J19 and sensory sensitivity (r_s_ = 0.378, *p* < 0.01) as well as sensation avoiding (r_s_ = 0.422, *p* < 0.01) of the AASP. There were significant correlations between the EOE (ease of excitation) of the HSPS-J19 and low registration (r_s_ = 0.358, *p* < 0.01), sensory sensitivity (r_s_ = 0.366, *p* < 0.01), and sensation avoiding (r_s_ = 0.333, *p* < 0.01; Table 4) of the AASP.

### 3.5. Comparison between Males and Females

There were no significant differences in the self-report rating scales and GSR between men and women.

## 4. Discussion

This study examined the relationship between sensitivity tendency and psychological stress reactivity using the GSR in healthy students.

### 4.1. Changes in GSR by Stroop Task

In both the congruent and incongruent conditions of the Stroop task, the changes in GSR significantly decreased. In addition, comparisons of the maximum rates of change in GSR between the congruent and incongruent conditions demonstrated significantly lower values in the incongruent condition than in the congruent condition (−33.68 ± 30.48%, *p* < 0.01 and −44.85 ± 29.85%, *p* < 0.01). Therefore, as reported in a previous study [45], the incongruent condition induced more stress responses than the congruent condition.

### 4.2. Relationships between Self-Report Rating Scales and GSR

There were no significant correlations between the GSR and the total score and each subscale score of the self-reported scale HSPS-J19 in the congruent and incongruent conditions. Moreover, there were no significant correlations between the AASP and GSR in the congruent and incongruent conditions; however, there was a marginal trend toward significance between low registration of the AASP and GSR in both the congruent and incongruent conditions. The lack of a significant relationship between individual differences in sensitivity on the self-reported rating scales and the GSR, which is a measure of stress reactivity, suggests that self-rated sensitivity was not directly reflected in physiological responses. This is in line with the previous finding that subjective emotional states do not necessarily correlate with physiological responses [46]. Furthermore, previous studies have reported that the change in skin potential level due to mental sweating is more pronounced when the task is more difficult [47] and that the change in skin conductance response when performing the Stroop task is smaller in young participants than in older participants [48]. The Stroop task used in this study was likely not very difficult for the healthy students. In other words, the response in GSR to the Stroop task may not have been sufficient to reflect the degree of self-reported rating scales.

On the other hand, low registration of the AASP tended toward a significant association with the changes in GSR during the Stroop task in this study (r_s_ = 0.231, *p* < 0.10; r_s_ = 0.219, *p* < 0.10), suggesting that participants with low registration in the AASP present a weak response to mental stress because of their lower sensitivity. This result is in line with previous findings [49] that people with high scores on low registration showed a lower stress response reflecting mental sweating. It can be postulated that the rating scale of the AASP tends to reflect the characteristics of participants who are less responsive to mental stress.

This study hypothesized that higher SPS would be associated with a greater objective sweating response; however, the results did not support this hypothesis. This contradictory study result may be due to the unique response characteristics of the Japanese participants in this study. Williams et al. argued that cultural differences in SPS are a subject for future research [50]. Taking into account the anthropological characteristics of the participants in the evaluation of SPS may be necessary.

### 4.3. Relationship between Self-Report Rating Scales

The HSPS-J19 total score and EOE had significant positive correlations with low registration (r_s_ = 0.323, *p* < 0.01; r_s_ = 0.358, *p* < 0.01), sensory sensitivity (r_s_ = 0.425, *p* < 0.01; r_s_ = 0.366, *p* < 0.01), and sensation avoiding (r_s_ = 0.434, *p* < 0.01; r_s_ = 0.333, *p* < 0.01) of AASP, respectively, and LST of HSPS-J19 had significant positive correlations with sensory sensitivity (r_s_ = 0.378, *p* < 0.01) and sensation avoiding (r_s_ = 0.422, *p* < 0.01) of the AASP. First, high SPS indicated by total HSPS-J19 scores may be considered contrary to the low registration of the AASP, which is characterized by low responsiveness to sensory stimuli due to high sensory thresholds; however, this low registration may indicate that the original hypersensitive system shut down [51]. Second, the sensory sensitivity of the AASP, which represents discomfort with sensory stimuli, can be regarded as conceptually similar to the high SPS indicated by total HSPS-J19 scores. Therefore, the significant correlations between total HSPS-J19 scores and sensory sensitivity and sensation avoiding of the AASP may mean that the behavioral pattern of minimizing or avoiding unpleasant stimuli was involved [7,52]. One study [27] examined the relationship between the total score of the short version of the HSPS [8], which is based on the original version of the HSPS [3], and the AASP. The study found significant correlations between the total score of the short version of the HSPS and low registration, sensory sensitivity, and sensation avoiding of the AASP, which was consistent with the results of the present study.

In addition, both LST and EOE of the HSPS-J19 represent low thresholds for sensory stimuli and are conceptually similar; therefore, their relationship reflects each other [53]. For this reason, some argue [28] that LST and EOE can be considered as one. The majority of the original version of the HSPS [3] reflects the negative aspects of stimuli [5], and LST and EOE are subscales that precisely correspond to these. One study [30] reported that the correlation patterns of the LST and EOE with other scales were similar to those with the HSPS total score; therefore, the present results were considered valid.

### 4.4. Comparisons by Gender

Although no significant differences were found between men and women in any of the comparisons, one study [54] demonstrated that GSR to visual stimuli was greater in men than in women. However, other studies reported no gender difference in temperament and personality [55] or HSPS [56]. In addition, there were no gender differences in the performance in the Stroop task [57], supporting the results of the present study, which found no gender differences in stress reactivity caused by the Stroop task either.

### 4.5. Limitations

The study included students at a single university; therefore, the generalizability of its findings should be carefully examined. In this regard, future studies must investigate participants of different generations and clarify the difference in reaction among generations. In addition, the Stroop task used as a stress task may not have been difficult for the healthy students in this study. A study [26] that examined the relationship between the HSPS original version and emotional responses demonstrated that the emotional responses in the brain when presented with a facial image of other persons expressing joyful or sorrowful emotions were greater in individuals with high SPS. Therefore, examining the relationship between the HSPS-J19 and changes in GSR when presented with facial images of others expressing various emotions may provide some new insights. Further studies are necessary to clarify these points.

## 5. Conclusions

This study examined the relationship between sensitivity tendency and psychological stress reactivity using the GSR in healthy students. It is one of the few studies to examine the relationship between SPS and physiological responses. Our results indicated that self-rated sensitivity might not necessarily be associated with an objective measure. Therefore, future studies should use sensitivity assessment scales other than the HSPS to examine SPS. Future studies must also use multiple indicators to evaluate the degree of SPS.

## Figures and Tables

**Table 1 healthcare-10-00949-t001:** Comparison of GSR between the congruent and incongruent conditions.

Variable	Congruent Condition	Incongruent Condition	*p*-Value
GSR	−33.68 ± 30.48	−44.85 ± 29.85	<0.001

Wilcoxon signed-rank test, mean ± SD. GSR = Galvanic skin response.

**Table 2 healthcare-10-00949-t002:** Correlation between HSPS-J19 and maximum rate of GSR change.

	HSPS-J19	LST	EOE
GSR in the congruent condition	−0.036	−0.090	0.007
GSR in the incongruent condition	0.033	0.022	0.010

Spearman’s rank correlation coefficient. HSPS-J19 = total score of the Japanese version of the Highly Sensitive Person Scale. LST = Low sensory threshold. EOE = Ease of excitation. GSR = Galvanic skin response.

**Table 3 healthcare-10-00949-t003:** Correlation between AASP and maximum rate of GSR change.

	Low Registration	Sensation Seeking	Sensory Sensitivity	Sensation Avoidant
GSR in the congruent condition	0.231 ^†^	−0.183	0.000	−0.017
GSR in the incongruent condition	0.219 ^†^	−0.079	0.009	0.152

Spearman’s rank correlation coefficient. ^†^ *p* < 0.10. AASP = Adolescent/Adult Sensory Profile. GSR = Galvanic skin response.

**Table 4 healthcare-10-00949-t004:** Correlation between HSPS-J19 and AASP.

	Low Registration	Sensation Seeking	Sensory Sensitivity	Sensation Avoidant
HSPS-J19	0.323 **	−0.044	0.425 **	0.434 **
LST	0.185	−0.039	0.378 **	0.422 **
EOE	0.358 **	−0.080	0.366 **	0.333 **

Spearman’s rank correlation coefficient. ** *p* < 0.01. HSPS-J19 = total score of the Japanese version of the Highly Sensitive Person Scale. LST = Low Sensory Threshold. EOE = Ease of Excitation. AASP = Adolescent/Adult Sensory Profile.

## Data Availability

All data generated or analyzed during this study are included in this published article.

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
