# Peer review of "Relationship between Sensitivity Tendency and Psychological Stress Reactivity in Healthy Students"

_healthcare, 2022, doi:10.3390/healthcare10050949_

Round 1

Reviewer 1 Report

Thank you for allowing me to review this manuscript. This manuscript entitled "Relationship between Tendency to Sensitivity and Psychological Reactivity 2 to Stress in Healthy Students".

It is an interesting and highly relevant article today, although it has several limitations that make it suitable for publication in this journal. These limitations are detailed below:

- In the introduction, it would be important to clearly state the objective of the study

- In the material and methods section it is described in detail. However, as a signal and weakness, we can highlight that different important ethical considerations are not reflected. It would be necessary to indicate if the study has the authorization of the ethics committee. In addition, it would be important to highlight the inclusion and exclusion criteria of this research

- The results are presented in a clear and orderly manner. Also, an interpretation of them is reflected. However, in the tables the acronyms used in them are not specified at the foot of the table.

- A section that gives the article quality, being a strength of it, is the discussion. In the manuscript, the results are discussed in an orderly manner, with necessary citations and rigor.

- In relation to the conclusions, they are clear and precise. However, we recommend indicating more precisely the implication in clinical practice of the same. Also, it would be interesting to state the lines of the future that the authors consider.

- In relation to the conclusions, they are clear and precise. However, we recommend pointing out more precisely their implication in clinical practice and the line of future proposed.

  • Finally, in relation to the bibliographical references, the article shows a sufficient number of consulted articles, which supposes an important strength of the manuscript. However, as stated, the central theme of the manuscript is highly topical, which requires that the bibliography be from recent years. However, in the bibliographical references section, there are articles with more than 10 years. It would be necessary to review this aspect and eliminate those citations that are not up to date.
  • Nice job

Reviewer 2 Report

I will begin by stating that I applaud the theme of this study “Relationship between Sensitivity Tendency and Psychological Stress Reactivity in Healthy Students”. The manuscript presents a good study, with important findings. However, authors must improve it so that it has the scientific quality to be published.

Some improvement suggestions are indicated in the attached document.

Round 2

Reviewer 2 Report

Small changes must be included. Changes are indicated in the PDF.
